# Time to Revise the WHO Categories for Severe Rabies Virus Exposures–Category IV?

**DOI:** 10.3390/v14051111

**Published:** 2022-05-22

**Authors:** Stephen J. Scholand, Beatriz P. Quiambao, Charles E. Rupprecht

**Affiliations:** 1College of Medicine, University of Arizona, Tucson, AZ 85724, USA; 2Research Institute for Tropical Medicine, Alabang, Muntinlupa City 1781, Metro Manila, Philippines; bpquiambao@yahoo.com; 3LYSSA LLC, Atlanta, GA 30333, USA; charleserupprechtii@gmail.com

**Keywords:** immune globulins, lyssavirus, neglected diseases, prophylaxis, rabies, vaccines, zoonosis

## Abstract

Rabies is a devastating disease and affects millions of people globally, yet it is preventable with appropriate and timely postexposure prophylaxis (PEP). The current WHO exposure categories (Categories I, II, and III) need revision, with a special Category IV for severe exposures. Rare cases of PEP failure have occurred in severe bites to the head and neck. Multiple factors, including route, wound severity, depth, contamination, viral dose, proximity to highly innervated areas and the CNS, and the number of lesions, remain unconsidered. Injuries in areas of high neural density are the most significant considering lyssavirus pathophysiology. Current recommendations do not account for these factors. A Category IV designation would acknowledge the severity and the increased risk of progression. Subsequently, patient management would be optimized with wound care and the appropriate administration of rabies-immune globulin/monoclonal antibodies (RIG/MAbs). All Category IV exposures would be infiltrated with the full dose of intact RIG (i.e., human RIG or MAbs) if the patient was previously unvaccinated. More concentrated RIG/MAb formulations would be preferred. As a world rabies community, we cannot tolerate PEP failures. A fourth WHO categorization will improve the care of these high-risk patients and highlight the global health urgency of this neglected disease.

## 1. Introduction

Rabies, one of the most dreadful diseases of all time, disproportionally affects those in developing countries who experience significant health disparities [1]. Globally, tens of thousands of preventable deaths occur annually [2]. Approximately forty percent of cases are children, who typically suffer a dog bite but are not given adequate preventive care [3].

Global health efforts have strengthened over the last few years to eliminate human deaths from rabies, through a strategic, all-inclusive approach: the United Against Rabies Collaboration [4]. This redoubling of efforts will ideally impact morbidity and mortality, with the goal of zero deaths by 2030 via enhanced mass dog vaccination and the application of modern human prophylaxis [5,6]. We believe a revision of the current WHO categories of rabies virus (RABV) exposure, with optimized management, works toward this goal.

### 1.1. RABV Pathophysiology

After an animal bite or other significant exposure, highly neurotropic RABV and other lyssaviruses infect peripheral nerves and transit to the brain [7]. Preexposure prophylaxis (PrEP) for populations at risk is highly effective but expensive, so more cost-effective implementation is desirable [8]. Postexposure prophylaxis (PEP) is highly effective when correctly implemented but requires keen medical judgment and excellent health care provisions [9,10]. Rare cases of PEP failure have occurred, especially in severe bites to the head and neck, despite an appropriate provision of care [11,12].

### 1.2. WHO Exposure Categories and PEP Protocols

The WHO categorizes RABV exposures based on relative risks, matched to a prophylaxis paradigm. This aids clinicians to utilize limited and expensive preventative resources most appropriately [13]. For the purposes of our discussion, we focus on RABV prophylaxis in the unvaccinated individual, exemplifying the majority of exposed patients.

Category I includes contact but non-RABV exposure, such as licks on intact skin or touching an animal determined to be rabid (without any break in the skin). Care is performed by washing of the affected areas without any need for a rabies vaccine or other biologics. Minor wounds such as a superficial scratch or abrasion without bleeding would be Category II. In these cases, localized wound care with lavage followed by the administration of a modern rabies vaccine series is recommended. Finally, Category III exposures are broad and include scratches with bleeding, mucosal contamination, plus all types of bites from rabid animals. The severity, number, and location of wounds are unaccounted for in this assessment. We believe that this is problematic, and is a major focus of our argument. Wound care, followed by rabies immune globulin (RIG) or monoclonal antibody (MAb) infiltration into and around the wounds, and rabies vaccination are recommended.

The main problem, particularly in developing countries, is access to RIG/MAbs, which are expensive and in limited supply. Considering the cost and availability of these efficacious products, the WHO has recently revised its rabies prophylaxis guidelines [13] and added equine products (i.e., ERIG) and MAbs to its Essential Medicine List [14]. Now, PEP includes a shortened intradermal vaccine series for Category II and III exposures. In addition, RIG/MAb usage, if available, is limited to infiltration of the wound sites only. Hence, depending on the nature of the injury, only a small amount of RIG/MAbs may be administered to a patient, for example, a wound on the tip of the nose or the ear, where anatomical space constrains volume infiltration and the effective delivery of RABV antibodies.

## 2. Proposal: Category IV

Thus, we propose that a fourth category be created for those exposures deemed the most critical. Our rationale includes multiple factors that require greater emphasis for this zoonosis. We believe one needs to consider wound characteristics including severity, number, and applied morphology with depth and tissue involvement, contamination, and proximity to highly innervated areas and the CNS. Injuries in areas of high neural density, such as the face, would be the most significant, considering lyssavirus pathophysiology.

Severe exposures to the head and neck (i.e., multiple, deep wounds), which are highly innervated areas and close to the brain, represent the greatest risk of progression to disease without proper intervention (Figure 1). The shortest incubation times in rabies occur in these settings and can be as little as one week after RABV exposure [15].

Currently, there are no ideal bite injury classification systems, which may partially explain why the current WHO categorization is underdeveloped in its current form and has persisted for decades without major revision.

The majority of RABV exposures are Category III but lack any refinement or triage of these exposures. For example, data from Tunisia showed that Category III comprised most of the exposures, including all bites, at 63.7% (*n* = 29,062). Category I was the least common, with licks/touching at 3.8% (*n* = 1466), and Category II consisting of scratches and abrasions without bleeding made up 21.2% (*n* = 8214) [17]. A focused analysis from China showed that for 711 human rabies cases, 63.3% had Category III exposures, 6.3% of the patients had Category I contacts, and 30.4% had Category II [18]. Another reported 564 of 1015 (55.6%) animal bite victims in China were Category III [19].

A data set of 422 human rabies cases from Bangladesh showed a greater preponderance of Category III exposures, with 95% occurring as bites, which were all grouped as Category III (*n* = 399). The remainder were mostly Category I, and only 23 were Category II (scratches) [15].

## 3. Would a Fourth Category Make a Difference?

True failures of PEP have occurred, although these are admittedly rare, considering that rabies has the highest case-fatality of any infectious disease [11,12,20].

The burden of true failures is unknown because of limited reporting and inadequate laboratory confirmation of human rabies cases from around the world.

Most failures of rabies prophylaxis resulted from not following proper guidelines, including a late start of prophylaxis, insufficient cleansing of the wound, total omission of RIG/MAb administration, or failure to inject RIG/MAb into all wound sites. Concurrent immunosuppressive conditions or drugs might also be a factor [21].

In Wilde’s review, at least eight true failures occurred, where apparently all the steps of PEP were correctly implemented [12]. The data showed that face and/or neck bites were involved in five of these eight, and two cases involved a finger. Half of the cases (four of eight) involved multiple bites, all of which were from dogs. In the majority of cases (five of eight) ERIG was used. Of interest, only three countries were included in that review, suggestive of reporting bias given the relatively robust rabies surveillance and public health infrastructures in those countries.

## 4. Other Considerations for Category IV Exposure

We think neuroanatomy is also relevant in consideration of Category IV exposures. Recently, Bharti et al. reported a rabies case in a child whose facial nerve was severed, with the postmortem findings of facial nerve dissection at the parotid gland, with noted pathology of swelling and edema of the nerve stump [20]. Such an exposure is quite serious. The face, head, and neck would appear the most important sites of concern. Older data suggest that the progression to rabies without adequate interventions is predictable in severe cases, especially after attacks by rabid wolves. For example, reported rates of progression in unvaccinated archival cases were: head, 50% to 80%; finger/hand, 15% to 40%; and legs, 3% to 10% [22].

Such outcomes based upon lyssavirus pathobiology, related in part to neuroanatomy including inoculation, attachment or uptake into nerves, and axonal transport to nerve cell bodies, etc., appear intuitive. Lyssaviruses utilize the mammalian CNS as a fundamental niche, and biomedical interventions require a rapid and thorough response in kind given the speed of tropism [23].

Therefore, we propose that Category IV exposures include all severe bites to the face, head, and/or neck. The treating clinician should distinguish Category IV over Category III exposures utilizing their best clinical judgment. For example, multiple bites and/or very severe bites elsewhere on the body might be upgraded to Category IV exposure. Other clinical factors may play an important role, such as an anatomically tight space that presents a challenge for RIG/MAb infiltration. Very rarely, bites to the finger (also highly innervated but constrained in deliverable RIG/MAb volume to minimize the probability of compartment syndrome) have progressed to rabies, despite appropriate PEP [11]. We think that in such instances, the clinician should be able to triage to a higher level of concern considering the patient pool and supply availability. To aid clinicians, we suggest a modification of the basic WHO recommendations, as shown in Table 1.

## 5. Thoughts on Immunization Effectiveness with a Renewed Focus on Passive Immunity

All proposed Category IV exposures should be infiltrated with the full dose of HRIG/MAbs computed at 20 IU/kg, if the patient had not been previously immunized. We advocate for a more concentrated HRIG/MAb product, if available. Currently, there are 150 and 300 IU/mL licensed products [24]. The latter concentration delivers twice as much RABV antibody to the affected area with the same RIG volume. Injection of the remainder HRIG/MAb dose into the nearest anatomically feasible area provides a sustained reservoir of immediate passive immunity. This might also include subcutaneous tissues proximal to the wound. For example, an injury on the eyelid might benefit from further HRIG/MAb injections subcutaneously into the forehead and/or cheek area as the anatomy may allow. Depot injection into a distal thigh muscle would seem comparatively less useful, as the localized diffusion of RABV antibody would be lower. Where HRIG/MAbs are not available, an additional dose of ERIG (40 IU/kg) on day four after the affected bite may be considered to bridge the gap before RABV-neutralizing antibodies appear actively after vaccination. We hypothesize this measure given the more rapid pharmacokinetic clearance of these equine products versus intact Ig biologics. Finally, these patients should be triaged, or given priority over lesser category exposure patients if limited HRIG/MAb supplies so dictate.

Differences exist in homologous human RIG versus heterologous equine products. Because of the inherent antigenic dissimilarity of a heterologous equine product, those molecules are treated to reduce reactogenicity (i.e., anaphylaxis and serum sickness), but at a cost in potential effectiveness. On a molecular level, the Fc portion of the equine IG is deleted, affecting its function in conformational support, as well as its role in antibody-dependent cellular cytotoxicity, resulting in a shorter half-life. Questions have been raised about the in vivo effectiveness related to such differences. For example, in an experimental PEP animal model, Schumacher et al. documented that the ED50 using F(ab’)2 fragments of a MAb was 34.0 IU compared with 2.4 IU for an intact MAb, suggesting the importance of the Fc region for greatest efficacy [25]. Similarly, Hanlon et al. found in a hamster model that there were significant differences in HRIG vs. treated groups given heterologous biologics, with more than 90% mortality for some formulations of equine product when used without vaccine [26]. We suggest that HRIG/MAb may need to be used preferentially in severe RABV exposures because of these potential concerns.

## 6. Summary Recommendations

Recognition of Category IV RABV exposures should motivate the clinician to provide the highest level of care possible. We think that this change will improve delivery in several ways. With regards to the patient, wound care must be optimized, and these patients should be prioritized to always receive RIG/MAbs. With regards to public health, we hope to influence transition to HRIG or MAb formulations over heterologous, nonintact equine products. Management protocols may be developed and/or enhanced to include a specially reserved supply of RIG/MAbs for triage of such higher-risk patients [16]. Understandably, resources are limited in many parts of the developing world, but we believe that there is a benefit in instituting a new standard to save lives, particularly in pediatric cases that are especially challenging [27].

Patients would continue to receive the previously recommended maximum 20 IU/kg dose. Where it is not feasible to obtain small, reserved supplies of HRIG, an additional dose of equine product (40 IU/kg) on day four to the affected area is a reasonable alternative. Of course, standard follow up care, including active immunization with rabies vaccines, should proceed.

Finally, while increased access to PrEP is laudable, we think that renewed attention to unpredictable, severe RABV exposures and proper patient care during PEP will positively impact this neglected disease, moving closer to the goal of zero dog-mediated human rabies deaths by 2030 in a One Health context [28].

## Figures and Tables

**Figure 1 viruses-14-01111-f001:**
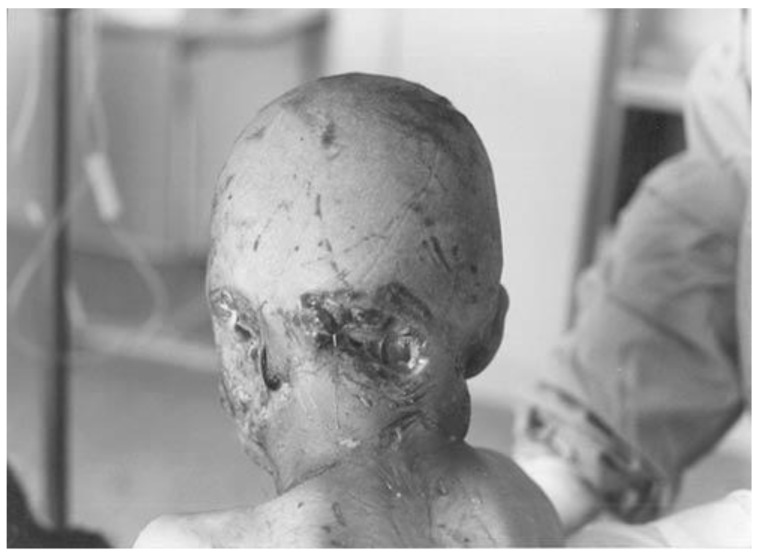
Young child savagely attacked by a rabid wolf, with severe bites to the ear and posterior scalp down to bone. An example of the proposed Category IV exposure, with kind permission from Dr. E Mostafavi [16].

**Table 1 viruses-14-01111-t001:** Revised WHO rabies exposure categories.

Category I: Touching or feeding animals, animal licks on intact skin (no exposure)
Category II: Nibbling of uncovered skin, minor scratches or abrasions without bleeding (exposure)
Category III: Single or multiple transdermal bites or scratches, contamination of mucous membrane or broken skin with saliva from animal licks, exposures due to direct contact with bats (severe exposure)
Category IV: Special consideration of severe bites to the face, head, and/or neck, or any additional considerations deemed by the clinician as extremely worrisome for RABV transmission (extremely severe exposure)

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
