# Peer review of "Time to Revise the WHO Categories for Severe Rabies Virus Exposures–Category IV?"

_viruses, 2022, doi:10.3390/v14051111_

Round 1

Reviewer 1 Report

Rabies is a zoonotic viral disease and is almost 100% fatal after onset of disease. However, the disease onset can be preventable by receiving of appropriate post-exposure prophylaxis (PEP). Currently, WHO classify the exposure into three categories (Category I, II, and III). Category III is severe exposures and requires administration of rabies immune globulin (RIG) or monoclonal antibody (mAb) infiltration into and around the wounds in addition to repeated vaccinations for appropriate PEP. Unfortunately, PEP was sometimes failed in the case of head, face and/or neck bite. Some nerves distributed in the face and neck have direct contact with the brain, through which rabies virus can quickly enter the brain from the peripheral tissues. Therefore, this review propose Category IV exposure, which includes all severe bites to the face, head and/or neck. In this case, the RIG/mAb infiltration must be provided, and the category designation make clinicians clear what they should do.

I strongly agree with their opinion and think that there is nothing to point out in this paper. I consider that the rationale for the need to establish Category IV are well described.

Author Response

Thank you very much for your thoughtful review!  We appreciate it greatly.

Stephen

Reviewer 2 Report

The authors proposed to revise the current WHO exposure category because of the rare cases of PEP failure in severe bites to the head and neck. They expressed concerns over the recent changes in WHO rabies prophylaxis guidelines with a shortened intradermal vaccine series and limited infiltration of the wound sites with RIG/Mab are not sufficient to protect against more serious exposures, especially the disease progression can be fast in some cases.  Therefore, category IV exposure was proposed for the severe cases to receive the maximum dose of 20 IU (hRIG) and 40 IU (eRIG) per kg body weight plus an additional on day 4 if eRIG was used. The concerns expressed by the authors are understandable; however, the following issues are identified.

  1. It is not clearly explained why a fourth category can make a difference. The WHO guidelines were drawn up to make the best use of biologics available in resource-limited countries and to save the maximum numbers of lives. In rich countries, hRIG is used in wider situations, e.g. CDC guideline states “people who have never been vaccinated against rabies previously, postexposure prophylaxis (PEP) should always include administration of both hRIG and rabies vaccine.” When the lack of resources is the main issue, an additional category IV will not change the situation.
  2. For the very severe bites of head areas, the authors have not cited the literature to show the recommended maximum hRIG treatment can make a difference.
  3. No literature is cited to justify the day 4 additional dose for eRIG, although it sounds reasonable.
  4. Line 225. The title of reference 10 is missing.

Author Response

Reviewer 2 Points and Counterpoints:

  1. POINT: It is not clearly explained why a fourth category can make a difference.

-COUNTERPOINT: It is true, that it might not make a difference at first, BUT, we believe it will ‘shine a light’ on the problems with the current framework and prepare the way for improved management of these cases. We believe that to create change, one has to lead – and we are no longer satisfied with the current paradigm. This is partly what should make our paper thought provoking and stimulatory; after this clarion call for change. we hope the status quo could change for the better. So to address this point, a few changes at the end of the manuscript have been made.

POINT: The WHO guidelines were drawn up to make the best use of biologics available in resource-limited countries and to save the maximum numbers of lives. In rich countries, hRIG is used in wider situations, e.g. CDC guideline states “people who have never been vaccinated against rabies previously, postexposure prophylaxis (PEP) should always include administration of both hRIG and rabies vaccine.” When the lack of resources is the main issue, an additional category IV will not change the situation.

COUNTERPOINT: In a similar vein, we want this to change – by proxy. Raising the bar on Rabies PEP could have a downstream effect on these other cases that are not managed properly. For too long Rabies has been neglected, and we feel more attention is due on this front. Doesn’t it seem so ‘blunt’ that all bites are lumped into one category? Surely we can do better…

  1. POINT: For the very severe bites of head areas, the authors have not cited the literature to show the recommended maximum hRIG treatment can make a difference.

COUNTERPOINT: This is a difficult issue, yes, and one that deserves more scientific discourse. Hopefully this paper would  start those conversations in earnest. There are suggestions that this Cat IV revision would make a difference – from Dr. Wilde’s paper, the majority of true failures were with eRIG, not an HRIG. In addition, higher dose formulations were not available at that time (so this would be a new area of research and investigation).

  1. POINT: No literature is cited to justify the day 4 additional dose for eRIG, although it sounds reasonable.

COUNTERPOINT: True, this is our hypothesis just put forward now in this review. We believe there is a rationale for this based on the more rapid metabolism of eRIGs compared to HRIG molecules.  eRIG molecules are quite different indeed than human proteins: structurally they are less robust, and pharmacokinetically – they are cleared more rapidly. So we have added a clarifying sentence to the manuscript to clarify.

  1. Line 225. The title of reference 10 is missing. Added - Thanks

Academic editor's comment: "The authors should consider adding a small table that summarizes the current WHO categories and proposed new IV category.  Added – Thanks

Table 1 – Revised WHO rabies exposure categories

Category I touching or feeding animals, animal licks on intact skin (no exposure);
Category II nibbling of uncovered skin, minor scratches or abrasions without bleeding (exposure);
Category III single or multiple transdermal bites or scratches, contamination of mucous membrane or broken skin with
saliva from animal licks, exposures due to direct contact with bats (severe exposure).

Category IV special consideration of severe* bites to the face, head and/or neck, or any additional considerations deemed by the clinician as extremely worrisome for rabies transmission (extremely severe exposure).

Round 2

Reviewer 2 Report

I agree with the authors that this article should be published to provoke discussion and drive up the standard of care. Fundamentally, it is the scarcity of resources that prevents the victims from receiving adequate care. The change of categorisation may help to draw attention to the issue and to ration the limited resources more effectively.